# Phylogenetic Relationships among TnpB-Containing Mobile Elements in Six Bacterial Species

**DOI:** 10.3390/genes14020523

**Published:** 2023-02-19

**Authors:** Yali Wang, Mengke Guo, Naisu Yang, Zhongxia Guan, Han Wu, Numan Ullah, Emmanuel Asare, Shasha Shi, Bo Gao, Chengyi Song

**Affiliations:** 1College of Animal Science and Technology, Yangzhou University, Yangzhou 225009, China; 2Department of Immunology, School of Medicine, Shenzhen University, Shenzhen 518060, China

**Keywords:** TnpB, transposable elements, *IS605*, *IS607*, *IS1341*

## Abstract

Some families of mobile elements in bacterial genomes encode not only a transposase but also an accessory TnpB gene. This gene has been shown to encode an RNA-guided DNA endonuclease, co-evolving with Y1 transposase and serine recombinase in mobile elements *IS605* and *IS607*. In this paper, we reveal the evolutionary relationships among TnpB-containing mobile elements (TCMEs) in well-assembled genomes of six bacterial species: *Bacillus cereus*, *Clostridioides difficile*, *Deinococcus radiodurans*, *Escherichia coli*, *Helicobacter pylori* and *Salmonella enterica*. In total, 9996 TCMEs were identified in 4594 genomes. They belonged to 39 different insertion sequences (ISs). Based on their genetic structures and sequence identities, the 39 TCMEs were classified into three main groups and six subgroups. According to our phylogenetic analysis, TnpBs include two main branches (TnpB-A and TnpB-B) and two minor branches (TnpB-C and TnpB-D). The key TnpB motifs and the associated Y1 and serine recombinases were highly conserved across species, even though their overall sequence identities were low. Substantial variation was observed for the rate of invasion across bacterial species and strains. Over 80% of the genomes of *B. cereus*, *C. difficile*, *D. radiodurans* and *E. coli* contained TCMEs; however, only 64% of the genomes of *H. pylori* and 44% of *S. enterica* genomes contained TCMEs. *IS605* showed the largest rate of invasion in these species, while *IS607* and *IS1341* had a relatively narrow distribution. Co-invasions of *IS605*, *IS607* and *IS1341* elements were observed in various genomes. The largest average copy number was observed for *IS605b* elements in *C. difficile*. The average copy numbers of most other TCMEs were smaller than four. Our findings have important implications for understanding the co-evolution of TnpB-containing mobile elements and their biological roles in host genome evolution.

## 1. Introduction

Insertion sequences (ISs) are the simplest mobile genetic elements and only contain the gene required for transposition and regulation. Notably, IS elements are widely spread in prokaryotes, and some families invade eukaryotes [1,2]. Furthermore, ISs play an essential role in shaping the genome’s evolution and its hosts’ adaptability [3]. The *IS605* and *IS607* are recognized as two distinct IS elements with a similar structural organization. They have two open reading frames (ORFA and ORFB) and encode two nuclease genes flanked with subterminal left end (LE) and right end (RE) palindromic elements. The ORFBs from *IS605* and *IS607* encode the TnpB nuclease, harboring a conserved RuvC motif. In contrast, the ORFAs of *IS605* and *IS607* encode substantially different nuclease genes belonging to the distinct superfamily of enzymes and do not harbor the classical “DDE” domain of IS transposases [4]. The ORFAs of *IS605* encode single-domain proteins containing a single catalytic tyrosine recombinase (called Y1) which belongs to the HuH enzyme superfamily, containing a conserved amino-acid triad composed of Histidine (H)-bulky hydrophobic residue (u)-Histidine (H) [4]. Additionally, the ORFAs of *IS607* encode a serine recombinase (SR) which carries a predicted helix-turn-helix DNA binding domain at the N-terminus and a catalytic domain at the C-terminus [5,6].

The transposition activity of *IS605* and *IS608* from *H. pylori* [7,8,9,10] and *ISDra2* from *D. radiodurans* have been characterized [11,12], and their transposition mechanism has been analyzed extensively [4]. The transposition of *IS605* members can be described as a “Peel and Paste” single-strand transposition mechanism [4]; this distinguishes them from classical ISs, which move via double-strand DNA intermediates. Although the mechanism for the SR of *IS607* is still obscure, the transposition of *IS607* has been detected in *E. coli* using a mating-out assay [13]. It is commonly accepted that the Y1 transposase of *IS605* plays a crucial role in its transposition and is sufficient to promote its mobility in vivo and in vitro. While TnpB, designated as the accessory gene of *IS605*, is not required for the transposition of either *IS608* or *ISDra2* in *E. coli* and *D. radiodurans* [8,13,14,15], it has been suggested that TnpBs might play a regulatory role in transposition in bacteria [11]. The main function of TnpBs remains largely unknown. In very recent studies, both evolutionary analysis and biochemical experiments have suggested that TnpBs are RNA-guided endonucleases [16,17].

At least three groups (*IS605*, *IS607*, and *IS1341*) of TnpB-containing mobile elements (TCMEs) have been recognized. *IS605* and *IS607* carry two nuclease-encoding genes flanked with LE and RE elements. Although *IS1341* displays fundamental differences in its organization, it only harbors one nuclease encoding gene (TnpB) flanked by the end elements of ISs [4]. However, it is still unclear whether members of *IS1341* are autonomous or decay copies from *IS605* or *IS607* elements. These groups are widely distributed in prokaryotes and have been reported in eukaryotes [4,18]. *IS605*, *IS607*, and *IS1341* have been detected in 71 species, 34 species, and 76 species, respectively, based on ISfinder database (http://www-is.biotoul.fr). The structure and activity of *IS605*, *IS606*, *ISHp608* and *ISHp609* from *Helicobacter pylori*, belonging to *IS605* groups, have been well defined [15,18,19,20]. However, the evolution landscape of TCMEs is still poorly understood.

Here, the evolution profiles, including diversity, structure organization, and copy number in the genome of TCMEs across six species including *Bacillus cereus* (*B. cereus*), *Clostridioides difficile* (*C. difficile*), *Deinococcus radiodurans* (*D. radiodurans*), *Escherichia coli* (*E. coli*), *Helicobacter pylori* (*H. pylori*), and *Salmonella enterica* (*S. enterica*), were systematically investigated using the available genomes of high assembly levels (chromosome and complete). These data provide a basic framework for future evolutionary study, and the characteristics describing each subgroup may help in further experimental studies.

## 2. Materials and Methods

### 2.1. TnpB-Containing Mobile Elements Mining

All known TnpB proteins were collected from the ISfinder (https://www-is.biotoul.fr/index.php) accessed on October 2022 (including TCMEs: *IS605*, *IS607*, and *IS1341*) and Pfma (http://pfam.xfam.org/ accessed on 6 October 2022) databases with access numbers of PF12323, PF07282, and PF01385. Sequence lengths of less than 200 aa or longer than 600 aa were discarded. In total, 21,733 sequences were obtained and submitted for clustering using the cd-hit program with an 80% identity; 4399 clusters were obtained. The conserved RuvC domains (about 120 aa) of these TnpB representative proteins from each cluster were aligned to the RuvC hidden Markov model (HMM) profile of TnpB (ORFB_*IS605*.hmm, PF01385), which was downloaded from the Pfma database (https://pfam.xfam.org 6 October 2022), via the hmmalign program of HMMER3 [21]. The sequences with truncated RuvC domains (greater than 20% gaps) were discarded. Finally, 3661 cluster-representative sequences with well-conserved RuvC domains of TnpB were obtained and submitted for rebuilding a new HMM profile of the TnpB-conserved domain with the hmmbuild program of HMMER3 [21]. The hmmsearch program of HMM3 was then used to search against the non-redundant protein sequences (NRs), which were downloaded from the NCBI database (https://ftp.ncbi.nlm.nih.gov/blast/db/FASTA/nr.gz, accessed on 6 October 2021), with the newly constructed HMM profile. Target hits (altogether 150, 497 proteins) with default parameters are reported. Four species (*Escherichia coli*, *Clostridioides difficile*, *Bacillus cereus*, *Salmonella enterica*) with the most abundant hits (Appendix A), and two species (*Deinococcus radiodurans* and *Helicobacter pylor*) with function at the TnpB-containing mobile elements previously defined by He [22,23] and Pasternak [11,14] were used for further TCME mining.

The genomes of a high assembly level (chromosome and complete) of *Bacillus cereus* (152), *Clostridioides difficile* (133), *Deinococcus radiodurans* (12), *Escherichia coli* (2467), *Helicobacter pylor* (335), and *Salmonella enterica* (1495) were downloaded from NCBI and used for TCMEs mining. These genomes were translated into proteins with the transeq program embedded in emboss software [24]. The TnpB homology proteins were then searched against the translated proteins by using the hmmsearch program of HMM3 with the HMM profile. Target hits with sequence lengths shorter than 50 aa were discarded, and the remaining target hits (genomic coordinates) were used to extract the DNA sequences with flank extensions of 1.7 kb upstream and 1.7 kb downstream of hits in genomes. The extracted DNA sequences were clustered by the USEARCH program with a 70% identity, and representative sequences from each cluster from each species were aligned by MAFFT. The boundaries of complete elements and structure organizations were manually checked and defined for each cluster. The elements with detectable LE and RE boundaries (identifiable cleavage site sequences) were designated full TCME elements. The translated proteins from the truncated elements with undetectable LE or RE or missing both LE and RE boundaries were used as queries to BLASTP against the ISfinder database to define their classification, and the truncated elements were discarded in cases of no homology hit to the ORFBs of *IS605*, *IS607,* or *IS1341* by BLASTP searching.

Finally, all the full TCME elements obtained were clustered again using the USEARCH program with a 90% identity. The consensus sequences were derived for each cluster and BLASTN to the ISfinder database, and the obtained IS elements were designated as the known IS elements; that is, if they shared over 90% sequence identity with any known sequences from *IS605*, *IS607,* or *IS1341* group with highly conserved LE and RE sequences and the cleavage sites. If otherwise, they were designated as new IS elements and named according to the nomenclature rules of ISfinder [25,26,27].

The upstream and downstream sequences of the TnpB ORFs of all the identified *IS1341* elements were aligned to the Y1 or SR ORF coding sequences (forward or reverse) of *IS605* and *IS607*. If the *IS1341* elements contained Y1 or SR-derived sequences (more than 20 bp match by alignment), they were designated as the decays of *IS605* or *IS607* and removed from the *IS1341* group.

### 2.2. Sequence Analysis and Phylogenetic Tree Construction

The potential ORFs of the obtained TCMEs were predicted by the BioEdit software. The protein domains were defined using the profile hidden Markov models by the online hmmscan web server (https://www.ebi.ac.uk/Tools/hmmer/search/hmmscan accessed on 6 October 2022). The obtained TnpA and TnpB sequences and the reference sequences from ISfinder (https://www-is.biotoul.fr/index.php accessed on 6 October 2022) were submitted for alignment using the E-INS-I method from the MAFFT software [28], and the final alignments were used for the phylogenetic tree construction. The phylogenetic tree was inferred [29], and the ultrafast bootstrap approach with 1000 replicates was applied. The best-suited aa substitution model was selected by ModelFinder.

### 2.3. Hairpin Structures Prediction

The repeat sequences at the subterminal were predicted by the Novopro web tool. The hairpin structures of LE and RE were predicted by the Mfold (DNA Folding Form (mfold.org, accessed on 6 October 2022) and RNAfold web servers (univie.ac.at, accessed on 6 October 2022).

## 3. Results

### 3.1. Distribution of TnpB-Containing Mobile Elements (TCMEs) in the Six Species of Bacteria

High-quality assembled genomes of six species, including 152 *B. cereus*, 133 *C. difficile*, 2467 *E. coli*, 12 *D. radiodurans*, 335 *H. pylori*, and 1495 *S. enterica* genomes, were downloaded and submitted for TCME annotation. Overall, 687 copies in *B. cereus*, 1875 copies in *C. difficile*, 44 in *D. radiodurans*, 4825 copies in *E. coli*, 734 copies in *H. pylori*, and 1831 copies in *S. enterica* were obtained and designated as TCME-derived sequences after filtering the sequences without any homology to the TnpBs of *IS605*, *IS605,* or *IS1341* from ISfinder, as previously described in the methodology.

The expansion of TCMEs across these species varied significantly: over 90% of *C. difficile*, *D. radiodurans*, and *E. coli* contained TCMEs, and 83% of the detected genomes of *B. cereus* harbored TCME copies. In comparison, only 64% and 44% of the detected genomes of *H. pylori* and *S. enterica* contained TCME copies, respectively. In addition, for TCME-detected genomes, the average copy number of TCMEs varied significantly across the six species (Table 1). The highest average TCME copy number (over 14) was observed for each genome of *C. difficile*, followed by *B. cereus* with an average of 5.28 TCME copies in each genome, and *D. radiodurans* and *H. pylori* with approximately 3.5 TCME copies in each genome. In contrast, the average copy numbers of TCMEs in *E. coli* (2.05 ± 0.94) and *S. enterica* (1.25 ± 0.61) were less than three in each genome. Most identified TCME copies (over 76%) were presented by full elements with detectable LE and RE boundaries (cleavage sites) in these species except for *S. enterica*, in which only 41% of TCME copies were full (Table 1).

### 3.2. Protein Type and Structure Organization of TCMEs in Six Species

Overall, 39 IS elements (including 27 new IS elements and 12 IS elements overlapping with ISfinder) were defined for all obtained TCMEs sequences in these genomes based on the structural organization and sequence identity summarized in Appendix A. The consensus sequences were derived for each IS element. Based on the structure organization, the 39 IS elements were classified into three groups (*IS605*, *IS607*, and *IS1341*), including 15 *IS605* elements, 6 *IS607* elements and 18 *IS1341* elements (Appendix A and Figure 1). Furthermore, different genetic structural organizations were observed for *IS605* (Figure 1A) and *IS607* (Figure 1B). The Y1, TnpB ORFs of *IS605a* and *IS605b* were oriented in the same direction with Y1 upstream of TnpB. However, the ORFs of *IS605a* partially overlapped. In contrast, *IS605b* was separated (Figure 1A). Similar structures were observed for *IS607a* and *IS607b* (Figure 1B), but the ORFA of *IS607* encodes aserine recombinase. In addition, the ORFs of *IS605c* were separated and oriented in the reverse direction (Figure 1A). Only one structure was observed for 18 *IS1341* elements (Figure 1C), and we found that more than half (10 elements) of the detected *IS1341* elements were shorter than *IS605* and *IS607*, ranging from 1252 bp to 1393 bp. However, 8 *IS1341* elements had relatively long lengths ranging from 1512 bp to 2382 bp. The lengths of *IS1341* TnpBs ranged from 362 aa to 489 aa (Appendix A).

Most *IS605* elements have a total length of 1.74~1.95 kilobases (kb), two ORFs that encode Y1 transposase of about 140 (102–164 aa), and TnpB nucleuses of about 400 (340–489) aa which are flanked by LE and RE and their cleavage sequences (Appendix A). Most *IS607* elements have a total length of 1.62~2.40 kb and harbor two ORFs that encode serine recombinases of about 200 (150–217 aa) and TnpB nucleuses of about 400 (361–447) aa, respectively, which are flanked by LE and RE and their cleavage sequences (Appendix A). Both left and right cleavage sites of these IS elements are relatively divergent across 39 IS elements (Appendix A). However, some cleavage motifs are more frequently detected, such as TCAA (16 IS elements) and TTCA (8 IS elements) for RE cleavage sites and TTAT (10 IS elements) for LE cleavage sites (Appendix A).

All elements of the *IS607* carry several flawed, short, directly repeated sequences at the left end (LE) and right end (RE) and show a lack of inverted repeat sequences at the subterminal (Appendix A) which were the putative binding sites of serine recombinases. The sub-terminal palindromic structures were detected for all LEs and REs of *IS605* elements (Appendix A), which were the putative binding sites of Y1 transposases, and the structures of hairpins were detectable for the LEs and REs of most *IS1341* elements. The alignment of the right flanks (the downstream of the coding sequences of TnpB to RE) of these TCMEs revealed that half of the identified IS elements (seventeen) contain two conserved (GC and G enriched) motifs in the right end of REs (Appendix A). In contrast, some TCMEs tend to increase AT and TG nucleotides in the right subterminal of REs (Appendix A), which may function as RNA guides.

### 3.3. Domains and Phylogenetic Analysis of Y1, SR, and TnpB

The overall sequence identity of 39 TnpBs was very low, representing about 24% and ranging from 7.2% to 98.7% (Figure 2A). The phylogenetic tree revealed that the mined TnpBs, together with the TnpB from ISfinder (233 proteins), were classified into two main branches (TnpB-A and TnpB-B) and two minor branches (TnpB-C and TnpB-D) with high bootstrap supports (>70%). IscBs, which were defined previously [19], were used as an outgroup (Figure 3A). In addition, the phylogeny of TnpB seems to be independent of the phylogeny of the TnpB-containing IS elements (Figure 3A). Here, we found that three RuvC (I, II, III) segments, two Zinc-fingers, an arginine/lysine-rich region (RK-rich) and a helix-turn-helix motif (HTH) embedded in the WED domain in the N-terminal, were identified in the alignments of TnpB (Figure 4A,B). In addition, all TnpB proteins, which are highly conserved (Appendix A), were defined by Karvelis et al. [16]. Their domain organizations differed from that of IscB, which includes three RuvC I, II, and III motifs, one arginine-rich region (R-rich), one HNH motif, one CXXC zinc finger, and an additional PLMP in the N-terminal IscB [17].

The overall sequence identity (29.49 ± 12.55%, ranging from 16.9% to 89.5%) of mined Y1s was also detected to be low (Figure 2). The alignment of the fifteen mined Y1 proteins (102–164 aa) of the *IS605* family revealed five signature sequence motifs, including the His-hydrophobic-His (HuH) motif required for metal ion binding [29]. The HuH triad, catalytic tyrosine (Y), and glutamine (Q) in the C terminal, which are important residues for the transposition [4,30], were highly conserved across these elements (Figure 4B). Most mined Y1s of *IS605* seem to be intact; however, two (ISEc46 and ISBce24) are truncated in the N terminal.

Three key domains of SR, including the N-terminal DNA binding domain (DBD), the long central catalytic domain harboring the catalytic serine, and the C-terminal HTH domain were identified based on the alignment (Figure 4C). The average of SR protein sequence identity is approximately 29.32 ± 4.93%, ranging from 20.9% to 39.7% (Figure 2C). All mined SRs of *IS607* seem to be complete except ISBce17, with a truncated N terminal, and the whole DBD missing. Phylogenetic trees of Y1s and SRs revealed that these Y1 and SR proteins appear to fall into separate clades. Their associated left and right cleavage sites were generally conserved in the same or close clades (Figure 3), which was also observed for the TnpB tree (Figure 3A).

### 3.4. Differential Distribution of TCMEs in the Genomes of Six Species

Our data revealed significant differential invasions of TCMEs across six species. Two groups of TCMEs were detected in three species, *E. coli* (*IS605* and *IS1341*), *H. pylori* (*IS605* and *IS607*), and *S. enterica* (*IS605* and *IS1341*); all three groups of TCMEs were detected in *B. cereus* and *C. difficile*, while only one group (*IS605*) was detected in *D. radiodurans*. In addition, overlapping invasions of *IS605* and *IS1341* elements were observed for 229 *E. coli* and 63 *S. enterica* genomes, respectively. Co-invasions of *IS605* and *IS607* were observed for 44 *H. pylori* genomes, and overlapping invasions of *IS605*, *IS607*, and *IS1341* were observed for 26 *B. cereus* and 15 *C. difficile* genomes, respectively (Figure 5). *IS605* displayed the most extensive invasions in the genomes of six bacteria species: over 53.95%, 99.25%, 100%, 90.88%, 52.84%, and 20.27% of the total detected genomes of *B. cereus*, *C. difficile*, *D. radiodurans*, *E. coli*, *H. pylori*, and *S. enterica* contained *IS605* copies, respectively, while *IS607* elements represent a relatively narrow distribution and were only detected in three species (*B. cereus*, *C. difficile*, and *H. pylori*). They also invaded into low proportions of these genomes, representing 31.58% of *B. cereus*, 11.28% of *C. difficile*, and 24.18% of *H. pylori*, respectively. *IS1341* was not detected in *D. radiodurans* and *H. pylori*, but presented in 112 *B. cereus*, 106 *C. difficile*, 312 *E. coli*, and 418 *S. enterica* genomes, accounting for 73.68%, 79.70%, 12.65%, and 27.96% of the total detected genomes of these species, respectively. Differential distributions of *IS605* and *IS607* elements were also observed across the investigated species (Table 2). The differential invasions of TCMEs in genomes may be related to the transposition activity of these elements; thus, the wide distribution of *IS605* elements indicates that they may possess a higher transposition activity than that of *IS607* and *IS1341* elements.

Substantial differential copy number variations were observed for the different groups and subgroups of TCMEs across these species. The average copy numbers of most TCME subgroups, including *IS605b*, *IS607a*, *IS607b* in *B. cereus*, *IS605a*, *IS607b*, and *IS1341* in *C. Diff*, *IS605c* and *IS1341* in *E. Coli*, *IS605a* in *H. pylori*, *IS605c,* and *IS1341* in *S. enterica*, range from one to two in each detected genome. Similarly, for each detected genome, approximately three copies of *IS1341* in *B. cereus*, *IS605a* in *D. radiodurans,* and *IS605c* and *IS607a* in *H. pylori* were observed. On average, over 12 copies were only observed for *IS605b* in *C. difficile* for each detected genome (Figure 6).

## 4. Discussion

### 4.1. Co-Evolution Profiles of TnpB, an RNA-Guided Endonuclease, with Y1 and SR

*IS605* was found by DNA sequencing as a putative, transposable element in a very early study [18], which was further confirmed in an *H. pylori* strain by hybridization and PCR [19]. Differential distribution of *IS605* in *H. pylori* isolates from different parts of the world was observed [30]. Later, the sequence diversity and copy numbers in five strains of *H. pylori* were estimated in Southern blots of genomic DNAs; furthermore, PCR and hybridization showed a differential distribution of *ISHp608* in *H. pylori* from different human populations were observed [15]. The invasion of *IS200/IS605* was recently characterized in *Halanaerobium hydrogeniformans* [31].

Recent research revealed that the RNA-guided endonucleases of IscB and TnpB were identified as the ancestral proteins of the Cas9 and Cas12 nucleases, respectively, and have adopted CRISPR–Cas systems [1,4,32]. Similarly, they display cleavage DNA activities in both prokaryotes and eukaryotes. TnpB is guided by an RNA derived from the right-end element of a transposon [16] to cleave DNA next to the transposon-associated motif (named TAM). In contrast, IscB is guided by a noncoding RNA (called OMEGA RNA) derived from the upstream of IscB ORF [17]. In the current study, by mining TnpB-containing mobile elements (TCMEs), we characterized the co-evolution profiles of TnpB endonuclease with Y1 and serine recombinases in six species of bacteria. In total, 9996 TCMEs were identified in 4594 genomes from six species, and they were classified into 39 IS elements based on their structure organization and identity, belonging to the three designated groups of *IS605*, *IS607*, and *IS1341* [4]. Our data mining revealed significant variations of TCME expansion across the genomes of these species and strains, and differential invasions of *IS605*, *IS607,* and *IS1341*, and their subgroups across these species and even across different strains. Furthermore, our data revealed that the phylogeny of TnpB is not related to the phylogeny of these mobile elements harboring TnpB proteins, which may indicate the existence of horizontal TnpB gene transfer among these groups of IS elements.

Surprisingly, we found most genomes of *C. difficile* (99%), *D. radiodurans* (100%), *E. coli* (94%), and *B. cereus* (83%) were invaded by TCMEs, while only about half of the *H. pylori* (64%) and *S. enterica* (44%) genomes contained TCME copies. Across the groups (*IS605*, *IS607*, and *IS1341*) of TCMEs, we found that the *IS605* represented the widest distribution in the detected species and invaded all six species. However, dramatic differences in *IS605* amplification in strain genomes between species were detected. *IS605* invaded only about 20% of *S. enterica* strain genomes, and about 50% of *B. cereus* and *H. pylori* strain genomes harbored *IS605* copies. In addition, most *C. difficile-*, *D. radiodurans-*, and *E. coli*-detected genomes (>90%) contained *IS605* elements. *IS607* and *IS1341* displayed a relatively narrow distribution and invaded into few species and had fewer detected strain genomes of these species (<30%) compared to *IS605.* The exception to this is *IS1341* in *B. cereus* and *C. difficile*, which invaded into above 70% of the detected strain genomes. These data indicate that TCMEs and their families experienced dramatically differential evolutionary histories across these bacteria species. They played roles in shaping the genome evolution of hosts and may contribute to bacterial genome plasticity [3].

Our data analysis also revealed that the copy numbers of TCME subgroups varied significantly in the detected genomes. Most subgroups in the detected genomes of most species were represented by one to two copies, while three to four copies in the detected genomes were observed for *IS1341* in *B. cereus*, *IS605a* in *D. radiodurans*, *IS605c* and *IS607a* in *H. pylori*. Approximately 13 copies in the detected genomes of *C. difficile* were observed for *IS605b* (Figure 6). The expansion difference of TCMEs, their groups (*IS605*, *IS607*, and *IS1341*) and subgroups (*IS605a*, *IS605b*, *IS605c*, *IS607a*, *IS607b*, and *IS1341*) between species and strain genomes may be dependent on both hosts and mobile elements. The autoregulation of mobile elements has been observed in both prokaryotes and eukaryotes [33,34,35]. It has been proved that although Y1 recombinase is sufficient to carry out the cleavage and joining steps of *IS605*, TnpB is not necessary for the transposition based on the two members of the *IS605* (*IS608* and *ISDra2*) functional analysis [33,34]. On the other hand, it may inhibit transposition activity [11]. Extensive invasion of *IS605* suggests that, compared with *IS607* and *IS1341*, *IS605* may have higher transposition activity, mainly contributed by Y1 recombinase. Further comparative evaluation of the transposition activities between Y1 and serine recombinases will help interpret these results.

### 4.2. Structure Organization of TCMEs

Based on the TCME mining, three types of structures were identified for *IS605* which were summarized in the previous review [4]: two genetic structure types (SR and TnpB genes oriented in the same direction with encoding regions overlapping or separating) were identified for *IS607* (Figure 1). However, more genetic structures of *IS607* (SR and TnpB genes oriented in the reverse direction) may also exist in prokaryotes; this requires further analysis. *IS1341* represents one genetic structure and only encodes TnpB flanked with LE and RE. We applied a stringent standard to exclude the putative decays of the *IS605* and *IS607* elements from *IS1341* elements, where the upstream or downstream TnpB of *IS1341*s containing Y1- or SR-derived sequences (more than 20 bp) were designated as the decays of *IS605* or *IS607* elements and excluded from *IS1341* group. We found that *IS1341* displayed a distinct invasion profile from *IS605* and *IS607*, such as the *IS1341* elements in *S. enterica*, where 418 genomes (about 28%) contain *IS1341* copies, while 303 genomes (about 20%) contain *IS605* elements and *IS1341* and *IS605* only co-exist in 63 genomes (about 4%). According to the present findings, the identified *IS1341* may evolve independently and may not be decayed from *IS605* or *IS607*. Systematic *IS1341* mining in prokaryote genomes can provide insight into their evolutionary history and origin. The transposition activity of *IS1341* is worth further verification and would provide more substantial evidence for their origins.

Our data initially revealed four distinct branches of TnpB. All motifs of TnpB, including three RuvC (I, II, III) segments, two ZFs, one RK-rich, and one HTH, have been previously defined [16] and were highly conserved across species. However, variations of the C-terminal were observed, and their function remains unknown. Diverse Cas12 subfamilies, including miniature Cas12 proteins (Cas12f), were reported [36,37]; however, their phylogenetic and association analyses with TnpB, which was designated as the ancestor of Cas12 [16], remain largely unknown. In addition, it has been suggested that TnpB may evolve from IscB [17]. Systematically, mining TnpB and Cas12 homology proteins in prokaryotes and viruses and extensively analyzing their evolutionary relationships with IscB will provide a broader insight into the evolutionary history of these RNA-guided nucleases.

## 5. Conclusions

The current study identified 39 TnpB-containing mobile elements (TCMEs) from the total 4594 genomes of six bacteria species. TCME groups and subgroups displayed significant differential evolution profiles (copy number in genome and structural organization) across the six bacterial species. Overall, more than 80% of the genomes of *B. cereus*, *C. difficile*, and *D. radiodurans*, *E. coli* harbor TCMEs, while only 64% of *H. pylori* and 44% of *S. enterica* genomes contain TCMEs. *IS605* displays the most extensive invasions in these genomes, while *IS607* and *IS1341* elements represent a relatively narrow distribution. Co-invasions of *IS605*, *IS607*, or IS1341 elements were observed for many genomes. Significant copy number variations were detected for the subgroups of TCMEs across these species. Additionally, although the key domains and motifs of TnpBs and their associated Y1 or serine recombinases are highly conserved, their overall sequence identities are low.

## Figures and Tables

**Figure 1 genes-14-00523-f001:**
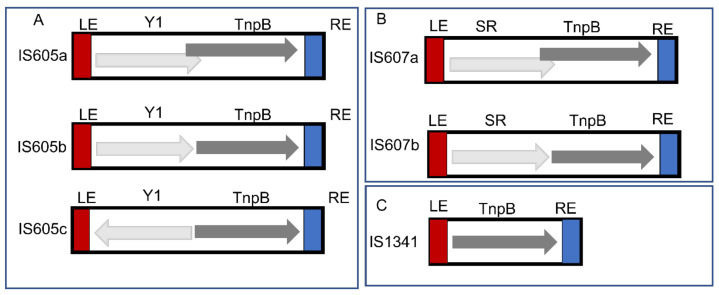
Structure organization of TCME elements. ORFs are presented as boxes with arrowheads showing the direction of transcription: LE and RE are red and blue boxes, respectively. (**A**) Three genetic structures of *IS605* elements. (**B**) Two genetic structures of *IS607* elements. (**C**) Genetic structure of *IS1341* elements.

**Figure 2 genes-14-00523-f002:**
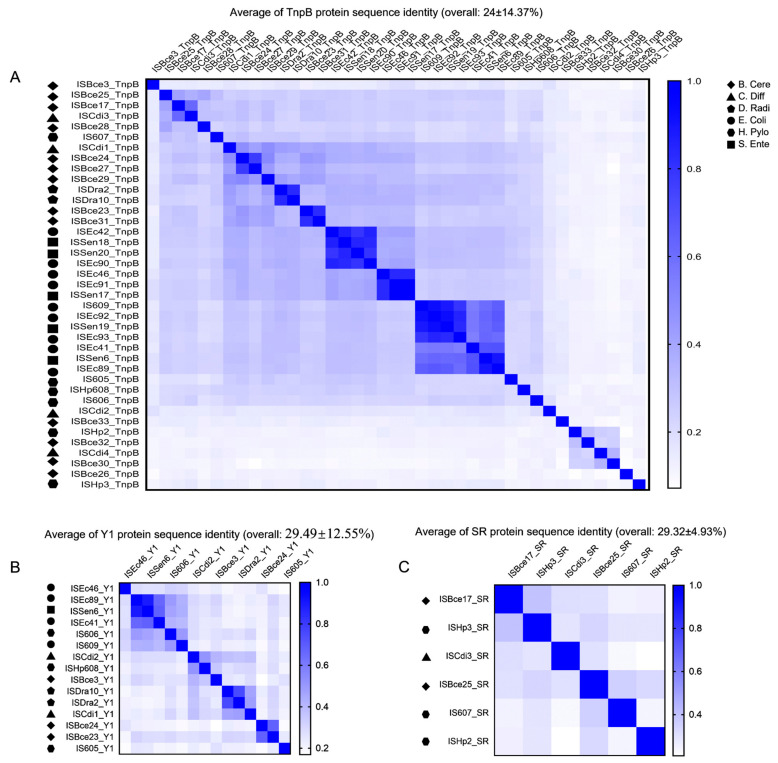
The overall identity of Y1, SR, and TnpB. (**A**) The identity of 39 mined TnpB proteins in six species. (**B**) The identity of 15 mined Y1 proteins in six species. (**C**) The identity of 6 mined SR proteins in six species.

**Figure 3 genes-14-00523-f003:**
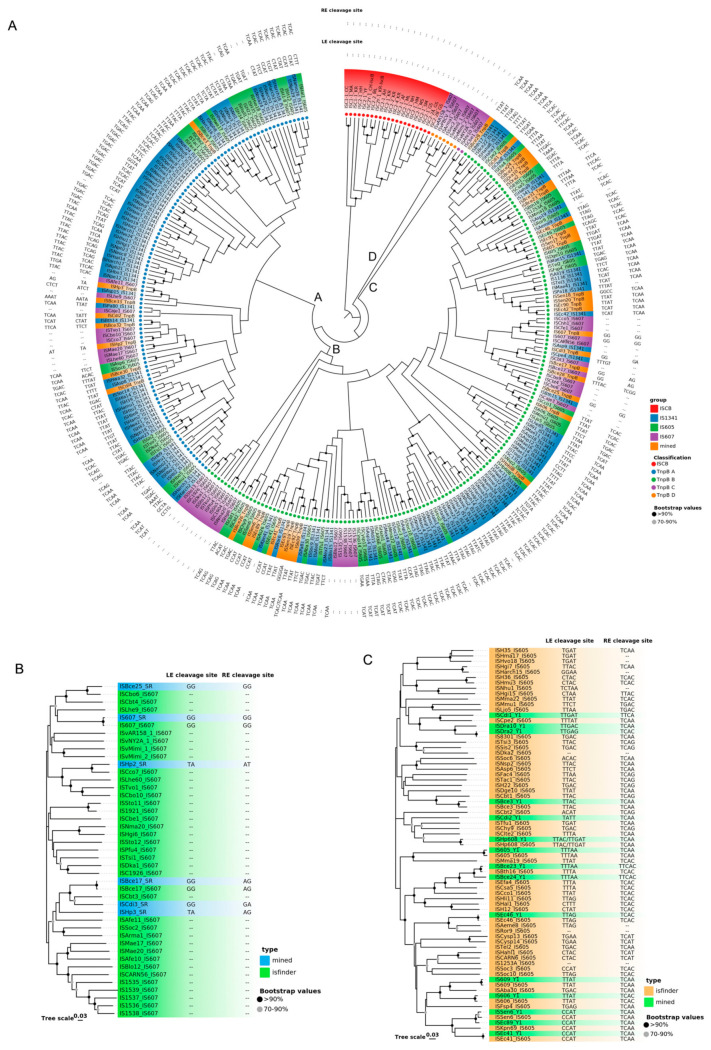
The phylogenetic tree of SR, Y1, and TnpB. The bootstrap values are indicated with a black circle (>90%) and a grey circle (70–90%). (**A**) The phylogenetic tree of TnpB and IscB. The TnpB proteins are classified into two main branches (TnpB-A and TnpB-B) and two minor branches (TnpB-C and TnpB-D), are indicated with blue, green, purple, and orange circles, respectively. The IscB formed a single branch shown with red circles. The mined TnpBs from six species are also shown in orange background. The *IS1341* subgroup is shown in blue background. The *IS605* subgroup is shown in green background. The *IS607* family is shown with purple background. (**B**) The phylogenetic tree of SR. The SR proteins from ISfinder are shown in green background. The mined SR proteins are shown in blue background. (**C**) The phylogenetic tree of Y1. The Y1 proteins from ISfinder are shown in orange. The mined-Y1 proteins are shown in green.

**Figure 4 genes-14-00523-f004:**
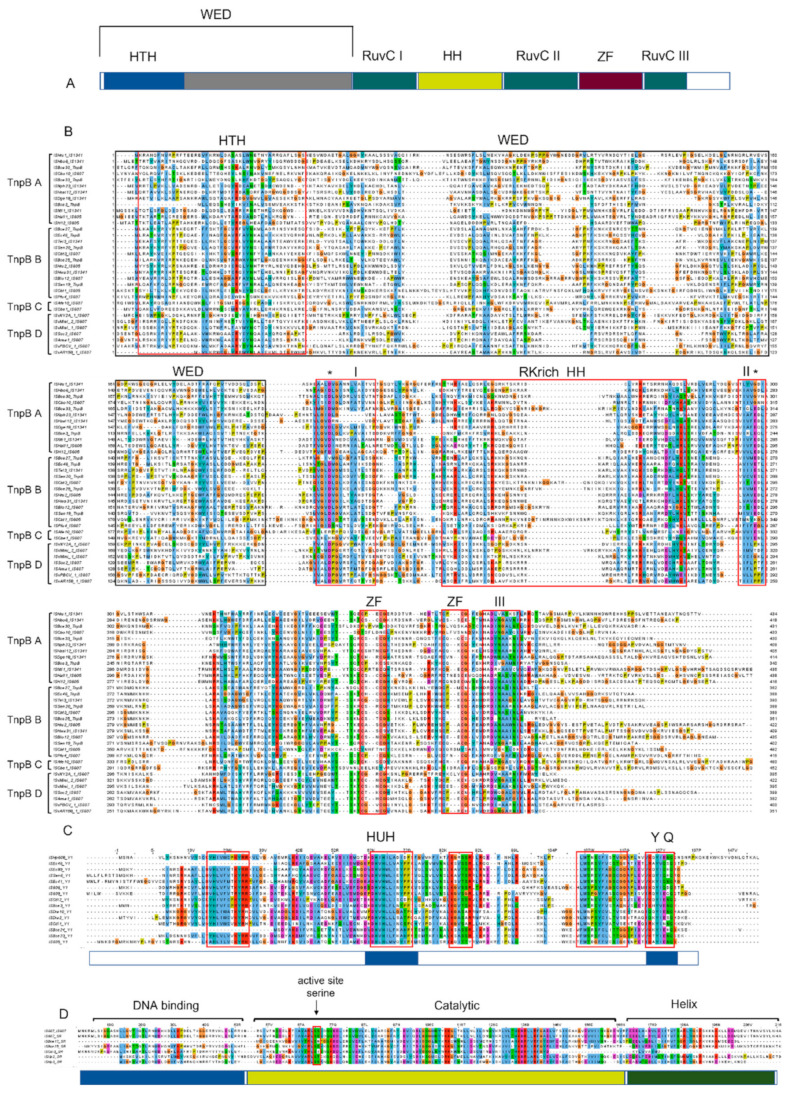
The alignments of TnpB, Y1, and SR protein sequences. The alignment was performed with MAFFT and drawn by Jalview Version 2. (**A**) Schematic of TnpB. The key domains include HTH—helix turn helix (blue), WED—wedge domain (grey), RuvC I, II, III (green), HH—arginine rich helix (yellow), ZF—zinc finger (red). (**B**) The alignment of several sequences was selected from TnpB-B, TnpB-B, TnpB-C and TnpB-D subgroups according to the phylogenetic tree. The HTH, RKrich HH, ZF, and RuvC I, II, and III domains are marked using a red box. (**C**) The alignments of the Y1 sequence. The HuH, Y and Q residues are indicated. (**D**) The alignments of SR. The DNA banding, catalytic, and Helix domains are also shown. The active site seine is marked using a black arrow.

**Figure 5 genes-14-00523-f005:**
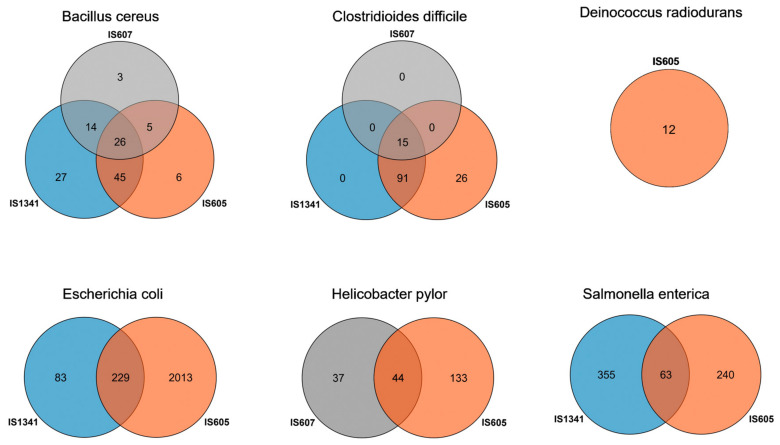
The genome number of different TCME groups in six species. The *IS607* is shown in grey color, *IS605* is shown in orange color, and the color blue represents *IS1341*.

**Figure 6 genes-14-00523-f006:**
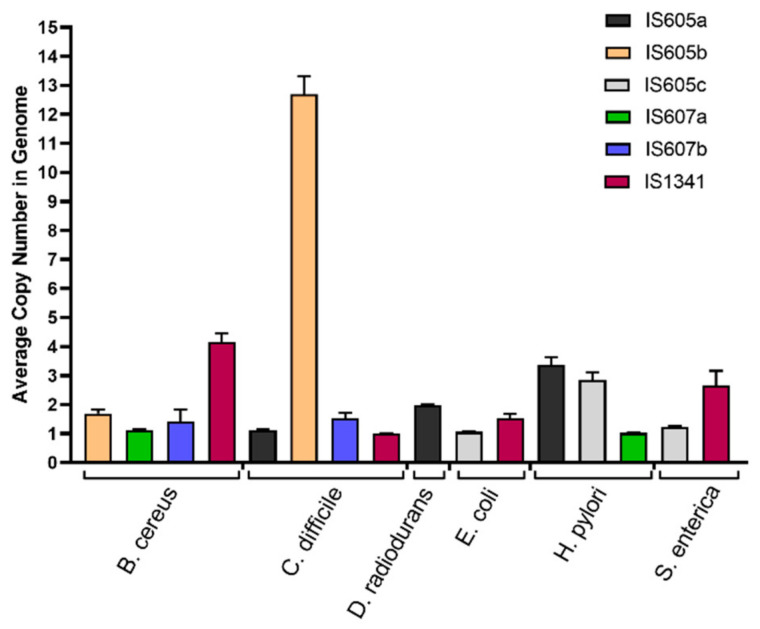
The average copies number of different subgroups of TCMEs in genomes of six studied bacterial species. The *IS605a*, *IS605b*, *IS605c*, *IS607a*, *IS607b*, and *IS1341* subgroups were shown in black, orange, grey, green, blue, and red, respectively.

**Table 1 genes-14-00523-t001:** Distribution of TnpB-containing mobile elements (TCMEs) in six species.

Species	Genomes Detected	Detected Elements	TCMEs	Genome Number Contain TCMEs/% ^a^	Average Copy Number of TCMEs in Each Genome	Copy Number of Full TCMEs/% ^b^
*B. cereus*	152	687	687	126/83%	5.28 ± 3.73	520/76%
*C. difficile*	133	1875	1875	132/99%	14.15 ± 6.82	1760/93%
*D. radiodurans*	12	52	44	12/100%	3.67 ± 1.65	44/100%
*E. coli*	2467	4837	4825	2325/94%	2.05 ± 0.94	4412/91%
*H. pylori*	335	735	734	214/64%	3.43 ± 2.99	678/92%
*S. enterica*	1495	1875	1831	658/44%	1.25 ± 0.61	757/41%

^a^ The genome number containing TCMEs accounts for the total detected genome number. ^b^ The copy number of full TCMEs accounts for the total copy number of TCMEs, Full TCME elements refer to elements with detectable LE and RE boundaries (cleavage sites).

**Table 2 genes-14-00523-t002:** Distribution of TCMEs in the genomes of six species.

Species	Total Genomes Detected	Genomes Number Containing *IS605*/% ^d^				Genomes Number Containing *IS607*/% ^d^			Genomes Contain *IS134* ^d^
		*IS605a*	*IS605b*	*IS605c*	Total	*IS607a*	*IS607b*	Total	*IS1341*
*B. cereus*	152	0	82/53.95	0	82/53.95	39/25.66	12/7.89	48/31.58	112/73.68
*C. difficile*	133	56/42.11	132/99.25	0	132/99.25	0	15/11.28	15/11.28	106/79.70
*D. radiodurans*	12	12/100	0	0	12/100	0	0	0	0
*E. coli*	2467	0	0	2242/90.88	2242/90.88	0	0	0	312/12.65
*H. pylori*	335	58/17.31	0	123/36.72	177/52.84	81/24.18	0	81/24.18	0
*S. enterica*	1495	0	0	303/20.27	303/20.27	0	0	0	418/27.96

^d^ The genome number containing TAMEs accounts for the total detected genome number.

## Data Availability

Not applicable.

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
