# Peer review of "Phylogenetic Relationships among TnpB-Containing Mobile Elements in Six Bacterial Species"

_genes, 2023, doi:10.3390/genes14020523_

Round 1
Reviewer 1 Report
Insertion sequences of IS200/IS605 and IS607 families encode tnpA transposase, which is essential for transposition, and often carry an accessory tnpB gene, which is dispensable for transposition. A bioinformatic analysis indicated that TnpB might be a predecessor of the CRISPR–Cas9/Cas12 nucleases. Therefore, I was very interested in the topic of the article. However, it turned out that in reality there is very little data in the article concerning the evolutionary aspects of the elements of this family. In fact, the main attention is paid to the distribution of the three groups of elements in the genomes of representatives of the 6 bacterial genera selected by the authors, as well as the structure of the studied elements. In order for the content of the study to be relevant to evolution, it would be necessary to describe the patterns of distribution of different groups\variants of ISs in different systematic groups. So far, the data obtained has not been analyzed for this purpose. Moreover, the novelty of the data obtained is not very clear. The results of other analyses are not even discussed in the Discussion.
Although the author has conducted an extensive bioinformatic analysis, and the topic is very intriguing, in this form the article looks incomprehensible and uninteresting. I will focus only on the main key points and considerations.
It is necessary to align the results of the work and the title of the article.
Introduction.
Please check the consistency of the presentation, paying special attention to the review of data on what is known to date and what aspects remain unclear, so that the goals and objectives of the presented research are clear.
Now the systematics of the elements containing tnpB is very incomprehensibly described. IS1341-type elements are part of the IS200/605 family. Based on gene composition, IS200/605 family can be classified into 3 distinct groups: (a) IS200 group, with only the tnpA gene; (b) IS605 group, where tnpA and tnpB are found adjacent and (c) IS1341 group, where tnpB is found alone (Gomes-Filho et al., 2015).
Methods
Please make sure that the methods are not described in other parts of the manuscript. For example, now there is a methodical fragment in the Discussion (lines 332-334). He could be here if the authors discussed the previously used methods gave unsatisfactory results, and thanks to the new approach they developed, new knowledge was gained.
The choice for the analysis of these 6 genera does not look very convincing. If you chose those genera where there were the most ISs or they were functionally well studied, then your data are of limited importance, both for studying the distribution of these elements during evolution and the evolution of their structure. This needs to be discussed. If you have received data on the distribution of ISs for all genomes in the Genbank, then it would be worth bringing them. Probably, some bacterial phylogenetic groups are simply poorly represented in the Genbank, and this, of course, is also a reason not to take them into account in the study, but it is necessary to worth noting.
Methods and their results
Please clearly describe step by step each analysis, its purpose, the results obtained and their novelty.
Now it is unclear from the text why there is such a different number of proteins in Figures 3 A, B and C and how sequences were selected and prepared for these analyses. In addition, Figure 3A looks mysterious to me. It would seem that each branch of proteins should correspond to anyone family/group of IS elements. And from the analysis it turned out that any type of proteins TnpB can be part of any IS element belonging to any family/group of IS elements. Only a group of ISC elements (they are designated that way by Kapitonov et al., 2016) and their proteins form one separate branch, and really IscB (insertion sequences Cas9-like OrfB) proteins are encoded in a distinct family of IS200/IS605 elements. Hence, TnpB phylogeny does not reflect the TAMEs phylogeny except for the ISC branch. These results are not discussed in the manuscript. But it is on the basis of phylogenetic analysis that the evolutionary history is well traced.
The co-evolution of proteins included in the IS elements is listed among the goals of the research. This is a very interesting task. But unfortunately, I didn't find this moment in the results of the work at all.
The text contains very long lists of digital values, which are visible from tables and figures. But there are practically no short intermediate conclusions that are so necessary.
When discussing the results, you need to compare your data with those previously obtained, drawing conclusions about the novelty of your results. Right now, in the Discussion, fragments of your data simply alternate with literary ones. Therefore, it is very difficult to assess the novelty and significance of this work.
Minor points
The abbreviation TAMEs (TnpB associated mobile elements) seems to me unfortunate. It may be confused with the well-known abbreviation TAM. And in English it is more correct to talk about TnpB-contained mobile elements. It is necessary that the text of the manuscript will be corrected by an English-speaking person, because now there are many poorly formulated phrases in the text.
When reading, I lacked a figure with a schematic domain organization of the analyzed proteins.
Author Response
Insertion sequences of IS200/IS605 and IS607 families encode tnpA transposase, which is essential for transposition, and often carry an accessory tnpB gene, which is dispensable for transposition. A bioinformatic analysis indicated that TnpB might be a predecessor of the CRISPR–Cas9/Cas12 nucleases. Therefore, I was very interested in the topic of the article. However, it turned out that in reality there is very little data in the article concerning the evolutionary aspects of the elements of this family. In fact, the main attention is paid to the distribution of the three groups of elements in the genomes of representatives of the 6 bacterial genera selected by the authors, as well as the structure of the studied elements. In order for the content of the study to be relevant to evolution, it would be necessary to describe the patterns of distribution of different groups\variants of ISs in different systematic groups. So far, the data obtained has not been analyzed for this purpose. Moreover, the novelty of the data obtained is not very clear. The results of other analyses are not even discussed in the Discussion.
Although the author has conducted an extensive bioinformatic analysis, and the topic is very intriguing, in this form the article looks incomprehensible and uninteresting. I will focus only on the main key points and considerations.
It is necessary to align the results of the work and the title of the article.
Thanks, we deleted the RNA-guided endonuclease in title to avoid confusing.
In the current study, 4594 well-assembled genomes from six bacteria species were used for annotation of the TnpB contained mobile elements (TCME). We systematically annotated these elements, and characterized the evolution landscapes of these TCMEs, and we found that TCME families and subfamilies displayed significant differential evolution profiles across the six genomes species. Overall, more than 80% genomes of B. Cere, C. Diff, and D. radi, E. coli harbor TCMEs, while only 64% of H. pylo and 44% of S. Ente genomes contain TCMEs. IS605 displays most extensive invasions in these genomes, while IS607 and IS1341 represent a relatively narrow distribution. Co-invasions of IS605, IS607, or IS1341 were observed for many genomes. Significant variations of copy number were detected for the families and subfamilies of TCMEs across these species. Additionally, we also defined the phylogenetic classification, structure organization and sequence identities, and domain conservation across these proteins (TnpBs and their associated Y1 or SR recombinases).
Our data extensively revealed the evolution of TnpB contained mobile elements in these 4594 well-assembled genomes, and the co-evolution (distribution and genomic copy variation) of TnpBs with their associated Y1 or SR recombinases (IS605 or IS607), which may help for understanding the evolutionary behavior of TnpB.
Introduction.
Please check the consistency of the presentation, paying special attention to the review of data on what is known to date and what aspects remain unclear, so that the goals and objectives of the presented research are clear.
Thanks, we have added previous studies in the start of discussion, and highlighted our findings.
Now the systematics of the elements containing tnpB is very incomprehensibly described. IS1341-type elements are part of the IS200/605 family. Based on gene composition, IS200/605 family can be classified into 3 distinct groups: (a) IS200 group, with only the tnpA gene; (b) IS605 group, where tnpA and tnpB are found adjacent and (c) IS1341 group, where tnpB is found alone (Gomes-Filho et al., 2015).
The IS200 is not in our scope, because it does not contain TnpB.
Methods
Please make sure that the methods are not described in other parts of the manuscript. For example, now there is a methodical fragment in the Discussion (lines 332-334). He could be here if the authors discussed the previously used methods gave unsatisfactory results, and thanks to the new approach they developed, new knowledge was gained.
We removed the repeats of describing.
The choice for the analysis of these 6 genera does not look very convincing. If you chose those genera where there were the most ISs or they were functionally well studied, then your data are of limited importance, both for studying the distribution of these elements during evolution and the evolution of their structure. This needs to be discussed. If you have received data on the distribution of ISs for all genomes in the Genbank, then it would be worth bringing them. Probably, some bacterial phylogenetic groups are simply poorly represented in the Genbank, and this, of course, is also a reason not to take them into account in the study, but it is necessary to worth noting.
We agree that analysis based on all prokaryotes (totally, 484, 510 genomes), including 11, 290 Archaea and 473, 220 Bacteria genomes, will greatly improve the importance of this study. However, it is a huge task to annotate about 500 k genomes, it is impossible to finish in short time. In addition, some poor assembled genomes may impact the accuracy of annotation of TCME. Thus, as we stated clearly in manuscript, our analysis focus on the four species (Escherichia coli, Clostridioides difficile, Bacillus cereus, Salmonella enterica), which display the most abundant hits of TnpB, and two species (Deinococcus radiodurans, and Helicobacter pylor), functions at the TnpB associated mobile elements previously defined by He et al., and Pasternak et al., were used for further TCME mining.
Overall, 152 Bacillus cereus, 133 Clostridioides difficile, 12 Deinococcus radiodurans, 2467 Escherichia coli, 335 Helicobacter pylor, and 1495 Salmonella enterica stain genomes were used, and all these genomes were well-assembled and the evolution profiles of TCMEs annotated by using these data are more convincing.
Methods and their results
Please clearly describe step by step each analysis, its purpose, the results obtained and their novelty.
Now it is unclear from the text why there is such a different number of proteins in Figures 3 A, B and C and how sequences were selected and prepared for these analyses. In addition, Figure 3A looks mysterious to me. It would seem that each branch of proteins should correspond to anyone family/group of IS elements. And from the analysis it turned out that any type of proteins TnpB can be part of any IS element belonging to any family/group of IS elements. Only a group of ISC elements (they are designated that way by Kapitonov et al., 2016) and their proteins form one separate branch, and really IscB (insertion sequences Cas9-like OrfB) proteins are encoded in a distinct family of IS200/IS605 elements. Hence, TnpB phylogeny does not reflect the TAMEs phylogeny except for the ISC branch. These results are not discussed in the manuscript. But it is on the basis of phylogenetic analysis that the evolutionary history is well traced.
The phylogenetic analysis included 39 mined TnpBs and 233 TnpBs from the ISfinder database, which has been stated in text (line 226-228).
Thanks, yes, it is. Actually, only the IscB elements are distinct from the TnpB branches, while TnpB from different groups (IS605, IS607, and IS1341) an their subgroups randomly distribute in tree, indicating that “it seems that the phylogenetic classification of TnpB has nothing to do with the classification based on the structure organization of their associated mobile elements (Figure 1), TnpB from different groups (IS605, IS607, and IS1341) can present in different branches (Figure 3A).”
We have added in results and discussion.
The co-evolution of proteins included in the IS elements is listed among the goals of the research. This is a very interesting task. But unfortunately, I didn't find this moment in the results of the work at all.
Thanks, IS605s are elements containing both of TnpB and Y1, while IS607 are elements harboring both of TnpB and SR, which are both co-evolved mobile elements. In addition, the Co-invasions of IS605, IS607, or IS1341 were detected in many genomes, they are also co-evolved events for these hosts.
The text contains very long lists of digital values, which are visible from tables and figures. But there are practically no short intermediate conclusions that are so necessary.
Thanks, we try to include more text to summarize the results.
When discussing the results, you need to compare your data with those previously obtained, drawing conclusions about the novelty of your results. Right now, in the Discussion, fragments of your data simply alternate with literary ones. Therefore, it is very difficult to assess the novelty and significance of this work.
Thanks, we included the main progress of previous reports and highlighted our main findings in conclusion.
Minor points
The abbreviation TAMEs (TnpB associated mobile elements) seems to me unfortunate. It may be confused with the well-known abbreviation TAM. And in English it is more correct to talk about TnpB-contained mobile elements. It is necessary that the text of the manuscript will be corrected by an English-speaking person, because now there are many poorly formulated phrases in the text.
Thanks, we replaced the TAME (TnpB associated mobile elements) with TCME (TnpB-contained mobile elements) in the whole MS, and the text has been extensively revised by a natively English-speaking scientist.
When reading, I lacked a figure with a schematic domain organization of the analyzed proteins.
We have included schematic domain organization for all analyzed proteins (TnpB, Y1 and SR) in Figure 4.
Reviewer 2 Report
Summary/finding of this paper
This submitted paper shows the distributions of TnpBs in six bacterial species, searching TnpBs homologs using 4594 complete/chromosome genome sequences. This study founds 39 homologous of TnpBs (Two types, IS200/IS606 and IS607). On the phylogenetic analysis, TnpBs were classified into mainly two branches A,B and showed the minor branches C,D. The key-motifs shown in TnpBs found in all branches seem to be highly conserved as they specifically encode RuvC domains. In their findings, they reported that TnpB associated mobile elements shows different orientations of their associated Y1 and serine recombinases. Also, copy numbers per genome of each bacterium also reveals that those elements were redundant in C. difficile.
However, I am concerned that the substantial parts of their findings were already reported previously ( Kapitonov et al., 2010, Siguier et al., 2014). This paper could be more informative for readers if the authors could show more detailed structure organizations of TAME elements.
Comments
-Figure 1
This paper claims that they reported structure organization of TAME element of subgroups of well-known mobile elements such as IS605. Mobile elements contain sub-terminal left end (LE) and right end (RE) palindromic elements. Recently, the important of 3 ‘prime end of tnpB (approximately 16 nt) followed by RE sequence and flanking ends were very import for functions of TnpB. This was the evidence that TnpB could function independent of their associated recombinases (Karvelis et al., 2021 and Altae-Tran et al., 2021). Conserved non-coding regions immediately downstream of the coding sequences of TnpBs would function as RNA guides. Analyzing/focusing those regions can also be also useful. More detailed schematic representation could be more informative. (for example, is RE sequence found right next to the 3’end of TpnB? Is there any variations? )
-Figure 3
Previously, subgroups of IS605 and IS1341 were shown in Figure 1. Figure 3 shows there were 4 classifications of TnpB on the phylogenetic tree. It would be more useful to add subgroups so we know if TpnB in the same-subgroup were clustered together or not.
-Discussion/Conclusion
This paper specifically looks at TnpB families of IS200/IS605 and covers 6 bacteria species. However, the organization of the transposons of the IS605 superfamily were already reported clearly in Kapitonov et al., 2010, Siguier et al., 2014 where they investigated additionally Cas9 and IscB families too.
I think the stronger conclusions or new findings should be more highlighted in discussion.
Author Response
This submitted paper shows the distributions of TnpBs in six bacterial species, searching TnpBs homologs using 4594 complete/chromosome genome sequences. This study founds 39 homologous of TnpBs (Two types, IS200/IS606 and IS607). On the phylogenetic analysis, TnpBs were classified into mainly two branches A,B and showed the minor branches C,D. The key-motifs shown in TnpBs found in all branches seem to be highly conserved as they specifically encode RuvC domains. In their findings, they reported that TnpB associated mobile elements shows different orientations of their associated Y1 and serine recombinases. Also, copy numbers per genome of each bacterium also reveals that those elements were redundant in C. difficile.
However, I am concerned that the substantial parts of their findings were already reported previously ( Kapitonov et al., 2010, Siguier et al., 2014). This paper could be more informative for readers if the authors could show more detailed structure organizations of TAME elements.
Reply 1
Thanks, we carefully checked the references by Siguier et al., 2014., and it is a review presented by Siguier et al., 2014. It did summarize some published information of IS605/200, however, they didn’t contribute to the substantial annotation of TCMEs in genomes. We also found several papers published by Siguier et al. et al., and we carefully checked the references by Kapitonov et al., which characterized the Cas9 homology protein, IscB, but we didn’t find our working is substantially overlapping with their data. The main progress of these studies includes differential distribution in different isolates or H. pylori stains from different human populations, and the copy number in five stains, which is stated in discussion section, they are substantial different from our data analysis. Our analysis included the well-assembled genomes from 152 Bacillus cereus, 133 Clostridioides difficile, 12 Deinococcus radiodurans, 2467 Escherichia coli, 335 Helicobacter pylor, and 1495 Salmonella enterica stains. We defined their genomic distribution of TnpB-containing mobile elements across these strains, and copy number variation between genomes. We characterized their structure organization, classification, and phylogenetic relationships of TnpB, Y1 and SR, and sequence identities, and etc. Even in Helicobacter pylor, where IS605 has been extensively reported, We also provide systematic annotation of TnpB-containing mobile elements (genomic variation across strains, families and subfamilies diversity), which are largely missing in early studies.
Overall, our main findings and data are not covered by these previous reports.
Comments
-Figure 1
This paper claims that they reported structure organization of TAME element of subgroups of well-known mobile elements such as IS605. Mobile elements contain sub-terminal left end (LE) and right end (RE) palindromic elements. Recently, the important of 3 ‘prime end of tnpB (approximately 16 nt) followed by RE sequence and flanking ends were very import for functions of TnpB. This was the evidence that TnpB could function independent of their associated recombinases (Karvelis et al., 2021 and Altae-Tran et al., 2021). Conserved non-coding regions immediately downstream of the coding sequences of TnpBs would function as RNA guides. Analyzing/focusing those regions can also be also useful. More detailed schematic representation could be more informative. (for example, is RE sequence found right next to the 3’end of TpnB? Is there any variations?).
Thanks, we have carefully checked the downstream sequences (100bp) of TnpB and to the RE, we did find some conservation motifs in right end of Res, and added in text
The alignment of the right flanks (the downstream of the coding sequences of TnpB to RE) of these TCMEs revealed half of the identified IS elements (seventeen) contain two conserved (GC and G enriched) motifs in the right end of REs, while some TCMEs tend to enrich Ts, which may function as RNA guides.
-Figure 3
Previously, subgroups of IS605 and IS1341 were shown in Figure 1. Figure 3 shows there were 4 classifications of TnpB on the phylogenetic tree. It would be more useful to add subgroups so we know if TpnB in the same-subgroup were clustered together or not.
Thanks, we have included the text in results.
“The phylogenetic position of TnpB seems to be independent of the classification based on its structural organization, and TnpB from different groups (IS605, IS607, and IS1341) can present in different branches (Figure 3A).”
-Discussion/Conclusion
This paper specifically looks at TnpB families of IS200/IS605 and covers 6 bacteria species. However, the organization of the transposons of the IS605 superfamily were already reported clearly in Kapitonov et al., 2010, Siguier et al., 2014 where they investigated additionally Cas9 and IscB families too.
Thanks, here, we agree that the organization of the transposons of the IS605 superfamily were reported, however, we present more detail information, and explicit the phylogenetic relationships, classification, their domain conservation, alignment, sequence identity, and etc. which are absent in previous studies. Please also refer to Reply 1.
I think the stronger conclusions or new findings should be more highlighted in discussion.
Thanks for reminding, we have reworded the conclusion and highlighted the new findings in discussion and conclusion.
Round 2
Reviewer 1 Report
The authors took into account some of the comments, and now the results of the work became more understandable. However, the text still contains many poorly formulated phrases, elementary errors and typos. There are no references to Figures 2B, 3BC, 4D, S3 in the text. The reference to Figure S4 appears later than the reference to S5.
The nomenclature of IS-elements is used very carelessly. Please, check throughout the text so that it is clear everywhere that IS 607 is a family, and IS605 and IS 1341 are groups from the IS200/IS605 family. Moreover, this should be clearly stated in the Introduction. The new elements you have discovered belong to these groups, but are not the IS607, IS605 or IS1341 elements themselves. The subgroups of IS-elements that you found (according to Figure 1) should be called subgroups everywhere, and not subfamilies or otherwise. Check and correct this throughout the text.
In Abstract you write that «We found that TCME families and subfamilies displayed significant differential evolution profiles across the six genomes species». - What do you mean by evolutionary profiles? What exactly did you show? This was unclear in the first version and it remains unclear.
English is still poor and therefore it is difficult to read the article. Here are just some incorrectly or poorly worded phrases:
L16-18. We identified 39 homologous of TnpBs which belong to two known families 16 (IS200/IS605 and IS607) were classified into different subgroups based on their configurations and 17 sequence identities.
L96-98. Then, the hmmsearch program of HMM3 was used to search against the non-redundant protein sequences (NR), which were downloaded from the NCBI BLAST databases database. – How was it done? You can search sequences in some database, but what can you search in the program? And NCBI BLAST is just a software package.
L118-121. The elements with detectable LE and RE boundaries (identifiable cleavage site sequences) were designated as full TCME elements, and the full TCME elements containing intact ORFA and ORFB for IS605 and IS607, or intact ORFB for IS1341, were designated as intact TCMEs - Here, you mean the related IS-elements, and not about the IS605, IS607 or IS1341 themselves, but it is unclear from the sentence.
L190-191. Furthermore, three subgroups of IS605 (Figure 1A) and two subgroups of IS607 (Figure 1B) in different configurations were detected. – This phrase confuses the reader, and its meaning becomes clear only from the following text. It should be clearly explained at once that the subgroups of elements differed in the mutual arrangement of the genes Y1 recombinase (transposase) or serine-site-specific recombinase SR and TnpB. And for me, the term “genetic structure” is more familiar and understandable than “configuration”.
L221. All members of the IS607 subgroup… - Which subgroup? Do you mean the IS607 family? You use the term subgroup for elements from the same group/family with different structures (Figure 1).
L230. …while some TCMEs tend to enrich Tc – Do you mean: “while some TCMEs tend to enrich behind TA”?
L239-241. In 239 addition, the phylogenetic position of TnpB seems to be independent of the classification based on its structural organization, and TnpB from different groups (IS605, IS607, and IS1341) can present in different branches – The sentence is incorrectly worded and misleads the reader. But this is a very important conclusion of your work. The phylogenetic position of TnpB cannot be unrelated to the structure of TnpB itself. If this were the case, it would mean that the phylogenetic analysis was done incorrectly. It follows from your data that the phylogeny of TnpB is not related to the phylogeny of the IS -elements that contain it, which may indicate the existence of horizontal tnpB genes transfer among these groups of IS elements. This should be clearly stated.
L301-303. IS 605 display most extensive invasions in these species, which detected in all species, over 53.95%, 99.25%, 100%, 90.88%, 52.84%, and 20.27% of the total detected genomes of B. cure, C. diff, D. radio, E. coli, H. pylo, S. ente contain IS605 copies, respectively.
L 390-392. IS200 transposase expression and transposition are aided by a small anti-sense RNA, the first palindrome, and the second stem-loop at LE, which is a small mobile element containing Y1 recombinase ORF flanked by LE and RE.
L400. …SR and TnpB ORFs expressed in the same direction with ORFs overlapping or separating…. - In addition to the fact that this is very bad English and therefore the meaning is difficult to understand, it is more correct to talk about genes, not about ORFs.
L424-426. Our data initially revealed that three distinct branches of TnpB which are (Figure 3), all motifs of TnpB, including three RuvC (I, II, III) segments, two ZFs, one RK-rich, and one THE previously defined by [16], were highly conserved across species.
If a native English-speaking person had read and corrected the manuscript, it would not have contained such phrases. Once again I urge the authors to submit the manuscript for editing. It is impossible to fix everything in the short time allotted to reviewers.
Typos
Please correct:
L167. IS finder instead ISfiner;
L197. for 18 instead for18;
L204. direction of transcription instead of direction of translation;
L369. In comparison instead In Comparison;
L390. antisense instead anti-sense;
L396. activity [11]. instead activity [11];
L437. TnpB- contained instead TnpB-associated.
Other notes
Figure 1. In the Figure 1, replace the word “group” with “group of IS elements”, “classification” to “protein type”, ISCB – to IscB. Figure 1 legend: IS605 is not a single IS-element, but a group of related IS-elements. This should be clearly stated. And the explanations in the legend are unnecessary. They are in the main text.
The name of Figure 6 should be corrected to: The average copies number of different subgroups of IS-elements in genomes of six studied bacterial species.
L141-142. Protein secondary structure predictions were performed using the PSIPRED program (http://bio- 142 inf.cs.ucl.ac.uk/psipred, accessed on 1st March 2021). – Where is the data of this analysis in the Results?
L157-180. It is not necessary to completely duplicate all the information from the table in the text. Shorten this part. You can write something interesting that follows from your data. For example, the fact that TCMEs are more common in gram-positive bacteria and there are more copies of them in the genomes than in gram-negative ones. Although to confirm this finding additional studies are needed. And this may be related to the evolutionary history of these two large phylogenetic groups of bacteria.
L294-324. It would be nice to at least slightly reduced the enumeration of data that can be seen in the figures and insert more generalizations or some conclusions. Well, for example, it can be seen from your data that the maximum number and variety of IS-elements were found in B. cure and C. diff. It remains unclear whether this is also typical for other Firmicutes.
L294-295. Please explain what you mean by evolutionary profiles. In fact, you describe the distribution of different types of IS-elements among representatives of the 6 species of bacteria you have chosen.
L340. The evolution profile of IS200/IS605 was characterized in Halanaerobium hydrogeniformans recently[38].- In this article (38), the term "the evolution profile" is not used. And the article is really not about that.
L372-374. These data indicate that TCMEs and their families experienced dramatically differential evolutionary histories across these bacteria species, and they contribute to bacterial genome plasticity and may impact the adaptability of their hosts [3]. – It is assumed that different IS-elements can contribute to bacterial genome plasticity. Work [3] is devoted to the review of various mechanisms of this contribution. The participation of the described TCMEs to bacterial genome plasticity and the adaptability of their hosts require separate evidences.
L375-379. Accordingly, this agrees with the study by Durrant et al. [39], after analyzing nine bacterial species genomes, and found that the mobile element repertoire and insertion rates vary across species, and integration sites often cluster near genes related to antibiotic resistance, virulence, and pathogenicity, indicating that mobile elements might affect antibiotic resistance, virulence, and pathogenicity of bacteria, and contribute to the microbe adaptation [39].- How does this fit in with your data? You haven't studied the association of you TCMEs with genes related to antibiotic resistance, virulence, and pathogenicity.
L415-424. I didn't understand how this information is related to the results of your work.
Author Response
The authors took into account some of the comments, and now the results of the work became more understandable. However, the text still contains many poorly formulated phrases, elementary errors and typos. There are no references to Figures 2B, 3BC, 4D, S3 in the text. The reference to Figure S4 appears later than the reference to S5.
Thanks, we have corrected these.
The nomenclature of IS-elements is used very carelessly. Please, check throughout the text so that it is clear everywhere that IS 607 is a family, and IS605 and IS 1341 are groups from the IS200/IS605 family. Moreover, this should be clearly stated in the Introduction. The new elements you have discovered belong to these groups, but are not the IS607, IS605 or IS1341 elements themselves. The subgroups of IS-elements that you found (according to Figure 1) should be called subgroups everywhere, and not subfamilies or otherwise. Check and correct this throughout the text.
Thanks, we have checked and corrected throughout the text.
In Abstract you write that «We found that TCME families and subfamilies displayed significant differential evolution profiles across the six genomes species». - What do you mean by evolutionary profiles? What exactly did you show? This was unclear in the first version and it remains unclear.
Thanks, we have reworded.
English is still poor and therefore it is difficult to read the article. Here are just some incorrectly or poorly worded phrases:
Thanks, we have carefully reworded and corrected the whole text.
L16-18. We identified 39 homologous of TnpBs which belong to two known families 16 (IS200/IS605 and IS607) were classified into different subgroups based on their configurations and sequence identities.
Thanks, we have corrected this sentence.
L96-98. Then, the hmmsearch program of HMM3 was used to search against the non-redundant protein sequences (NR), which were downloaded from the NCBI BLAST databases database. – How was it done? You can search sequences in some database, but what can you search in the program? And NCBI BLAST is just a software package.
Sorry for confusing, we have corrected.
L118-121. The elements with detectable LE and RE boundaries (identifiable cleavage site sequences) were designated as full TCME elements, and the full TCME elements containing intact ORFA and ORFB for IS605 and IS607, or intact ORFB for IS1341, were designated as intact TCMEs - Here, you mean the related IS-elements, and not about the IS605, IS607 or IS1341 themselves, but it is unclear from the sentence.
Sorry, we have deleted the confusing statement.
L190-191. Furthermore, three subgroups of IS605 (Figure 1A) and two subgroups of IS607 (Figure 1B) in different configurations were detected. – This phrase confuses the reader, and its meaning becomes clear only from the following text. It should be clearly explained at once that the subgroups of elements differed in the mutual arrangement of the genes Y1 recombinase (transposase) or serine-site-specific recombinase SR and TnpB. And for me, the term “genetic structure” is more familiar and understandable than “configuration”.
Thanks, we have reworded the text.
L221. All members of the IS607 subgroup… - Which subgroup? Do you mean the IS607 family? You use the term subgroup for elements from the same group/family with different structures (Figure 1).
Sorry for confusing, we have reworded.
L230. …while some TCMEs tend to enrich Tc – Do you mean: “while some TCMEs tend to enrich behind TA”?
Yes, we have corrected, here, we mean that some TCME contain GC and G enriched motifs in the right end of REs, while some TCMEs tend to enrich AT and TG in the right ends of REs
L239-241. In 239 addition, the phylogenetic position of TnpB seems to be independent of the classification based on its structural organization, and TnpB from different groups (IS605, IS607, and IS1341) can present in different branches – The sentence is incorrectly worded and misleads the reader. But this is a very important conclusion of your work. The phylogenetic position of TnpB cannot be unrelated to the structure of TnpB itself. If this were the case, it would mean that the phylogenetic analysis was done incorrectly. It follows from your data that the phylogeny of TnpB is not related to the phylogeny of the IS -elements that contain it, which may indicate the existence of horizontal tnpB genes transfer among these groups of IS elements. This should be clearly stated.
Thanks, we have reworded in results and discussion, and added clear statement.
L301-303. IS 605 display most extensive invasions in these species, which detected in all species, over 53.95%, 99.25%, 100%, 90.88%, 52.84%, and 20.27% of the total detected genomes of B. cure, C. diff, D. radio, E. coli, H. pylo, S. ente contain IS605 copies, respectively.
Thanks, we have reworded.
L 390-392. IS200 transposase expression and transposition are aided by a small anti-sense RNA, the first palindrome, and the second stem-loop at LE, which is a small mobile element containing Y1 recombinase ORF flanked by LE and RE.
Thanks, we have deleted.
L400. …SR and TnpB ORFs expressed in the same direction with ORFs overlapping or separating…. - In addition to the fact that this is very bad English and therefore the meaning is difficult to understand, it is more correct to talk about genes, not about ORFs.
Thanks, we have reworded.
L424-426. Our data initially revealed that three distinct branches of TnpB which are (Figure 3), all motifs of TnpB, including three RuvC (I, II, III) segments, two ZFs, one RK-rich, and one THE previously defined by [16], were highly conserved across species.
Thanks, we have reworded.
If a native English-speaking person had read and corrected the manuscript, it would not have contained such phrases. Once again I urge the authors to submit the manuscript for editing. It is impossible to fix everything in the short time allotted to reviewers.
Sorry, the whole text was edited a new native English-speaking person.
Typos
Please correct:
L167. IS finder instead ISfiner;
Corrected
L197. for 18 instead for18;
Corrected
L204. direction of transcription instead of direction of translation;
Corrected
L369. In comparison instead In Comparison;
Corrected
L390. antisense instead anti-sense;
Deleted
L396. activity [11]. instead activity [11];
Corrected
L437. TnpB- contained instead TnpB-associated.
Corrected
Other notes
Figure 1. In the Figure 1, replace the word “group” with “group of IS elements”, “classification” to “protein type”, ISCB – to IscB. Figure 1 legend: IS605 is not a single IS-element, but a group of related IS-elements. This should be clearly stated. And the explanations in the legend are unnecessary. They are in the main text.
Corrected
The name of Figure 6 should be corrected to: The average copies number of different subgroups of IS-elements in genomes of six studied bacterial species.
Corrected
L141-142. Protein secondary structure predictions were performed using the PSIPRED program (http://bio- 142 inf.cs.ucl.ac.uk/psipred, accessed on 1st March 2021). – Where is the data of this analysis in the Results?
Sorry, we have deleted the text, which is very early analysis, and we didn’t present the results in the final manuscript.
L157-180. It is not necessary to completely duplicate all the information from the table in the text. Shorten this part. You can write something interesting that follows from your data. For example, the fact that TCMEs are more common in gram-positive bacteria and there are more copies of them in the genomes than in gram-negative ones. Although to confirm this finding additional studies are needed. And this may be related to the evolutionary history of these two large phylogenetic groups of bacteria.
We have shorted this section and reworded.
L294-324. It would be nice to at least slightly reduced the enumeration of data that can be seen in the figures and insert more generalizations or some conclusions. Well, for example, it can be seen from your data that the maximum number and variety of IS-elements were found in B. cure and C. diff. It remains unclear whether this is also typical for other Firmicutes.
Thanks, the distribution difference may be associated with the transposition activity of the element, thus IS605 may represent high transposition activity, and more frequently mobile in genomes. We reworded in results and discussion. We lack data from other Firmicutes, and can’t draw a general conclusion.
L294-295. Please explain what you mean by evolutionary profiles. In fact, you describe the distribution of different types of IS-elements among representatives of the 6 species of bacteria you have chosen.
We replaced the evolution profile with “invasions”.
L340. The evolution profile of IS200/IS605 was characterized in Halanaerobium hydrogeniformans recently[38].- In this article (38), the term "the evolution profile" is not used. And the article is really not about that.
We reworded.
L372-374. These data indicate that TCMEs and their families experienced dramatically differential evolutionary histories across these bacteria species, and they contribute to bacterial genome plasticity and may impact the adaptability of their hosts [3]. – It is assumed that different IS-elements can contribute to bacterial genome plasticity. Work [3] is devoted to the review of various mechanisms of this contribution. The participation of the described TCMEs to bacterial genome plasticity and the adaptability of their hosts require separate evidences.
We have reworded as ”These data indicate that TCMEs and their families experienced dramatically differential evolutionary histories across these bacteria species, and they play roles in shaping the genome evolution of hosts, and may contribute to bacterial genome plasticity”
L375-379. Accordingly, this agrees with the study by Durrant et al. [39], after analyzing nine bacterial species genomes, and found that the mobile element repertoire and insertion rates vary across species, and integration sites often cluster near genes related to antibiotic resistance, virulence, and pathogenicity, indicating that mobile elements might affect antibiotic resistance, virulence, and pathogenicity of bacteria, and contribute to the microbe adaptation [39].- How does this fit in with your data? You haven't studied the association of you TCMEs with genes related to antibiotic resistance, virulence, and pathogenicity.
Deleted
L415-424. I didn't understand how this information is related to the results of your work.
Deleted
Reviewer 2 Report
tnpB has been extensively studied previously (both in vivo and in vitro).
The study claimed structural organisation, classification, and phylogenetic relationships of TnpB, Y1 and SR. The main finding is that TnpB seems to be independent of the classification based on its structural organisation and TnpB from different IS elements can be present in different phylogenetic branches. Still, the organiztions of TnpB were previosuly reported and studied.
To deliver new or more detailed information, at least, the author should present the genetic organisation of the gene loci more precisely. For example, Figure 1, the authors said that there are different configurations of TCMEs, but it is not very clear. Like previously reported papers (for example, Karvelis et al., 2021), it would be more informative if they labelled them properly. In figure 1, RE is a short repeat and its presence is now known to be importnat for the job of TnpB. But in the figure, it seems it is located far from the stop codon of the TnpB. This may confuse readers.
Author Response
tnpB has been extensively studied previously (both in vivo and in vitro).
The study claimed structural organisation, classification, and phylogenetic relationships of TnpB, Y1 and SR. The main finding is that TnpB seems to be independent of the classification based on its structural organisation and TnpB from different IS elements can be present in different phylogenetic branches. Still, the organiztions of TnpB were previosuly reported and studied.
We agree that TnpBs are RNA-guided endonucleases, their activity has been evaluated in vivo and in vitro, and the protein domains/motifs of TnpB also have been defined, however, the structure organization of TnpB associated mobile elements and their distributions in genomes, and phylogenetic classification of TnpB are largely unknown, which were addressed in this manuscript.
To deliver new or more detailed information, at least, the author should present the genetic organisation of the gene loci more precisely. For example, Figure 1, the authors said that there are different configurations of TCMEs, but it is not very clear. Like previously reported papers (for example, Karvelis et al., 2021), it would be more informative if they labelled them properly. In figure 1, RE is a short repeat and its presence is now known to be importnat for the job of TnpB. But in the figure, it seems it is located far from the stop codon of the TnpB. This may confuse readers.
Sorry for confusing, we redrew the Figure 1 for more accurate, and aim to show the different structure of TCMEs primarily in Figure 1.